# Efficient Quantum Monte Carlo simulations of highly frustrated magnets: the frustrated spin-1/2 ladder

Stefan Wessel[1], Bruce Normand[2], Frédéric Mila[3] and Andreas Honecker[4*]

**1** Institut für Theoretische Festkörperphysik, JARA-FIT and JARA-HPC,
RWTH Aachen University, D-52056 Aachen, Germany
**2** Laboratory for Neutron Scattering and Imaging, Paul Scherrer Institute,
CH-5232 Villigen-PSI, Switzerland
**3** Institute of Physics, Ecole Polytechnique Fédérale de Lausanne (EPFL),
CH-1015 Lausanne, Switzerland
**4** Laboratoire de Physique Théorique et Modélisation, CNRS UMR 8089,
Université de Cergy-Pontoise, F-95302 Cergy-Pontoise Cedex, France

⋆ andreas.honecker@u-cergy.fr

## Abstract

**Quantum Monte Carlo simulations provide one of the more powerful and versatile numerical approaches to condensed matter systems. However, their application to frustrated quantum spin models, in all relevant temperature regimes, is hamstrung by the infamous "sign problem." Here we exploit the fact that the sign problem is basis-dependent. Recent studies have shown that passing to a dimer (two-site) basis eliminates the sign problem completely for a fully frustrated spin model on the two-leg ladder. We generalize this result to all partially frustrated two-leg spin-1/2 ladders, meaning those where the diagonal and leg couplings take any antiferromagnetic values. We find that, although the sign problem does reappear, it remains remarkably mild throughout the entire phase diagram. We explain this result and apply it to perform efficient quantum Monte Carlo simulations of frustrated ladders, obtaining accurate results for thermodynamic quantities such as the magnetic specific heat and susceptibility of ladders up to $L = 200$ rungs (400 spins 1/2) and down to very low temperatures.**

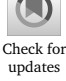

# 1 Introduction

Quantum Monte Carlo (QMC) simulation ranks among the most general and efficient methods for studying both the static and dynamic properties of quantum magnets at finite temperatures [1,2]. An early landmark example was provided by the square-lattice antiferromagnet, where field-theoretical predictions [3–5] were tested by QMC simulations for the spin-1/2 [6] and spin-1 cases [7]. A second valuable type of system was the $n$-leg spin ladder, which emerged as a tool for understanding the two-dimensional cuprates from the limit of one dimension [8–10], and where QMC simulations were essential for understanding the spin gap, correlation length, and magnetic susceptibility [11–15].

Geometrically frustrated magnets constitute an important class of quantum spin system with the potential to host exotic phases such as the quantum spin liquid [16–20]. However, QMC simulations on geometrically frustrated lattices suffer from the notorious "sign problem," which is the appearance of spin configurations with negative weights; a detailed discussion is deferred to Sec. 3. This problem restricts conventional QMC simulations to systems that are at most weakly frustrated [15, 21–23]. Although a general solution to the sign problem is not to be expected [24], progress has nevertheless been possible in some cases [25–32]. Specifically, for certain highly frustrated magnets, the Hamiltonian may be reexpressed in terms of cluster spins; if these form a bipartite lattice, sign-free QMC simulations are possible in the cluster basis [33–35]. For the example of a frustrated ladder, the clusters correspond to the ladder rungs. In the present manuscript, we will go away from the case of perfect frustration, where the sign problem can be eliminated completely [33, 34]. We will show that, although a sign problem remains present, it is so mild that the cluster basis allows efficient QMC simulations at all points in the phase diagram of the frustrated antiferromagnetic spin-1/2 ladder.

The structure of this article is as follows. We begin in Sec. 2 by presenting the model in detail. The QMC methods that we use to compute thermodynamic properties are introduced in Sec. 3, where the sign problem and its manifestations in the cluster basis are discussed in detail. Section 4 presents QMC results for the magnetic specific heat and susceptibility for sets of parameter choices representative of every region of the phase diagram (shown by the black dots in Fig. 1), which we compare with a range of numerical and theoretical results as an aid to

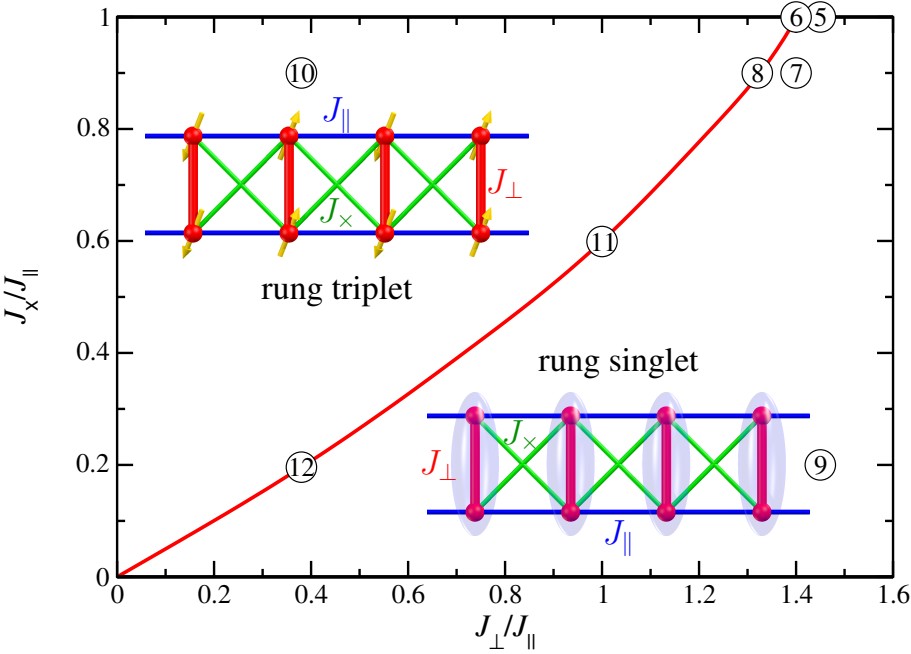

Figure 1: Ground-state phase diagram of the frustrated spin ladder, as established by a range of numerical methods [36–42]. The system shows only two phases, rung-singlet and rung-triplet, which are separated by a quantum phase transition (red line). In the inset schematics, the ladder sites (spheres) host $S = 1/2$ quantum spins and the Heisenberg couplings between spins are specified by the parameters $J_\perp$ for the ladder rungs, $J_\parallel$ for the ladder legs (blue), and $J_\times$ for the cross-plaquette couplings (green), which we take to be symmetrical. Purple rungs with ellipses represent rung-singlet spin states and red rungs with two parallel spins represent rung triplets. The numbered circles designate those points in the phase diagram for which we present thermodynamic results, each number matching that of the corresponding figure in Sec. 4.

physical interpretation. We summarize and offer some perspectives for reduced-sign-problem QMC in Sec. 5.

## 2  Model: Frustrated ladder

### 2.1  Hamiltonian and conservation laws

The Hamiltonian of a frustrated two-leg ladder with $L$ rungs, for any spin quantum number, $S$, is

$$H = J_\perp \sum_i \vec{S}_i^1 \cdot \vec{S}_i^2 + J_\parallel \sum_{i,m=1,2} \vec{S}_i^m \cdot \vec{S}_{i+1}^m + J_\times \sum_{i,m=1,2} \vec{S}_i^m \cdot \vec{S}_{i+1}^{\bar{m}}, \tag{1}$$

where $i$ is the rung index, $m = 1$ and 2 denote the two chains of the ladder, and $\bar{m}$ is the chain opposite to $m$. The superexchange parameters, $J_\perp$, $J_\parallel$, and $J_\times$ are depicted in the insets of Fig. 1 and we comment that the ladder we consider is always symmetrical under reflection through an axis bisecting all its rungs (i.e. under exchange of chains 1 and 2). In our numerical calculations we will impose periodic boundary conditions, such that $i + L \equiv i$.

Let us introduce the total-spin and spin-difference operators on rung $i$,

$$\vec{T}_i = \vec{S}_i^1 + \vec{S}_i^2, \qquad \vec{D}_i = \vec{S}_i^1 - \vec{S}_i^2. \tag{2}$$

The $SU(2)$ algebra of the operators $\vec{S}_i^m$ implies the on-site commutation relations

$$\left[T_i^\alpha, T_i^\beta\right] = \sum_{\gamma=x,y,z} \epsilon^{\alpha,\beta}{}_\gamma\, T_i^\gamma, \qquad \left[T_i^\alpha, D_i^\beta\right] = \sum_{\gamma=x,y,z} \epsilon^{\alpha,\beta}{}_\gamma\, D_i^\gamma, \qquad \left[D_i^\alpha, D_i^\beta\right] = \sum_{\gamma=x,y,z} \epsilon^{\alpha,\beta}{}_\gamma\, T_i^\gamma \quad (3)$$

for the operators of Eq. (2), where $\alpha, \beta = x, y, z$ are the Cartesian components of the spin operators and the commutators for different sites vanish automatically.

Using the composite operators (2), the Hamiltonian (1) can be reexpressed in the form

$$H = \sum_{i=1}^{L} \left( J_\perp \left( \frac{1}{2}\, \vec{T}_i^2 - S(S+1) \right) + \frac{J_\| + J_\times}{2}\, \vec{T}_i \cdot \vec{T}_{i+1} + \frac{J_\| - J_\times}{2}\, \vec{D}_i \cdot \vec{D}_{i+1} \right). \qquad (4)$$

We note that the exchange of leg and diagonal couplings, $J_\|$ and $J_\times$, yields an equivalent Hamiltonian: the first term of Eq. (4) is manifestly invariant and the second symmetric under the exchange of $J_\|$ and $J_\times$, while the last changes sign. However, this sign-change is easily compensated by exchanging the order of legs 1 and 2 on every second rung in the transformation (2). It follows that the system is symmetric under the interchange of $J_\|$ and $J_\times$ and therefore we restrict our considerations to the regime $J_\times \leq J_\|$.

At $J_\times = J_\|$, the final term in Eq. (4) disappears and the expression simplifies to [43, 44]

$$H = J_\| \sum_{i=1}^{L} \vec{T}_i \cdot \vec{T}_{i+1} + J_\perp \sum_{i=1}^{L} \left( \frac{1}{2}\, \vec{T}_i^2 - S(S+1) \right). \qquad (5)$$

At this fully frustrated point, the Hamiltonian (1) has $L$ purely local conservation laws, namely the total spin $\vec{T}_i^2$ on each individual rung, which may be encoded in additional quantum numbers, $T_i$. Although the form of Eq. (5) is valid for all $S$ [45], henceforth we consider exclusively the case $S = 1/2$, where $T_i$ takes the values 0 (a rung singlet) or 1 (a rung triplet).

## 2.2 Ground-state phase diagram

Before discussing the finite-temperature properties of the frustrated spin-1/2 ladder, it is useful to recall its ground-state phase diagram, which is shown in Fig. 1. Considering first the unfrustrated case, $J_\times = 0$, in the limit of strong rung coupling, $J_\perp \gg J_\|$, the system clearly adopts a gapped rung-singlet state [12, 46]. By contrast, for $J_\perp = 0 = J_\times$, one has decoupled spin-1/2 chains, which are known to be gapless [10, 47, 48]. While early numerical work suggested the possibility that a finite critical value of $J_\perp$ may be required to open the gap [11, 49], scaling and field-theory arguments led to the conclusions that the critical value vanishes and the gap scales linearly with $J_\perp > 0$ [50–52]. These results were confirmed by later numerical work [12, 13, 53] and the critical value $J_{\perp,c} = 0$ is now well established for $J_\times = 0$.

The other well-controlled case is the fully frustrated situation, $J_\times = J_\|$. The observations of the previous subsection explain the phase diagram along this line: the last term in Eq. (5) enforces all $T_i = 0$ for large $J_\perp$ (rung-singlet phase) while for small $J_\perp$ the fluctuations of the first term dominate, giving all $T_i = 1$ (rung-triplet phase) [43, 44, 54, 55]. It is firmly established that this is a direct first-order transition, and takes place at a critical coupling $J_{\perp,c} \simeq 1.401484 J_\|$ [43, 44, 54, 55], which is inferred from numerical results for the ground-state energy of the spin-1 chain [56, 57]. The first-order phase transition remains present for small deviations away from perfect frustration, $J_\times \neq J_\|$, and as such is easy to trace numerically in this regime [36, 37].

Returning to the limit of weakly coupled chains, $J_\perp, J_\times \ll J_\|$, a field-theoretical analysis [58, 59] predicted that the line separating these two phases approaches $J_\perp = 2J_\times$. In fact the field theory further predicts an intermediate columnar-dimer phase [59], although such a

phase has not been observed unambiguously in any numerical investigations of the antiferromagnetic frustrated ladder [36,37,60] despite repeated efforts to verify its existence [38–42]. By contrast, an intermediate columnar-dimer phase can be stabilized by ferromagnetic superexchange couplings [61] or by an additional next-nearest-neighbor coupling, $J_2$, along the ladder legs [40, 62–64]. Although the frustrated ladder has to date been studied primarily from a theoretical perspective, the recently synthesized compound $Li_2Cu_2O(SO_4)_2$ [65, 66] has a ladder geometry with $J_\times = J_\parallel$, although in this case $J_\perp$ is ferromagnetic and there is a significant antiferromagnetic $J_2$ [67].

In the following we will focus on the antiferromagnetic case with $J_2 = 0$. Figure 1 summarizes numerical results that have been obtained for the full phase diagram of the antiferromagnetic spin-1/2 ladder [36–42]. These interpolate the rung-triplet-to-rung-singlet quantum phase transition between the analytically known limits of a direct, first-order process at large $J_\perp$ to a complex process with the possibility of an invisibly narrow intermediate phase at small $J_\perp$.

## 3 Method: Quantum Monte Carlo simulations

### 3.1 Sign problem

Our aim is to use QMC simulations to compute thermodynamic quantities for the frustrated ladder. The standard approach for dealing with negative weights appearing for some configurations is to use a reweighting scheme that performs the QMC sampling with respect to their absolute values. However, this requires keeping track of the sign of each configuration and including it in any measurement, as discussed in Refs. [1, 24]. The performance of the simulation is then determined by the average sign, $\langle sign \rangle$: as long as this quantity is close to unity, Monte Carlo sampling is still efficient, whereas a small average sign must be compensated by a corresponding increase in the number of samples. Stated precisely, Monte Carlo errors decrease with the square root of the number of samples, and thus to compensate, for example, for an average sign of $10^{-2}$ it is necessary to run the code $10^4$ times longer.

Throughout most of the phase diagram of the frustrated ladder, QMC simulations of the Hamiltonian in the single-site basis (1) suffer from a severe sign problem. This was illustrated for a case on the fully frustrated line ($J_\times = J_\parallel$) in Fig. 8 of Ref. [33]. Here we present in Fig. 2 an illustration of the average sign for a less strongly frustrated case. We note that the scale of $\langle sign \rangle$ is logarithmic, and thus that there remains an exponential suppression both with decreasing temperature and with increasing system size [24]. For a system with $L = 200$ rungs, the average sign drops below $10^{-2}$ for $T \lesssim J_\parallel$, rendering the low-temperature region completely inaccessible for systems of any meaningful size.

While the structure and nature of the sign problem allow little hope that it can be overcome in general [24], its severity does still depend on the choice of basis. As noted in Sec. 1, frustrated quantum spin models have been identified where the sign problem can indeed be overcome, and their number continues to grow. Leading methods for tackling the sign problem to date include meron and nested cluster algorithms [27, 28, 30] and a suitable choice of simulation basis [25, 26, 29, 31–35, 68]. For the frustrated ladder, we follow the latter approach, taking the rung basis as a natural choice for the rewritten Hamiltonian (4).

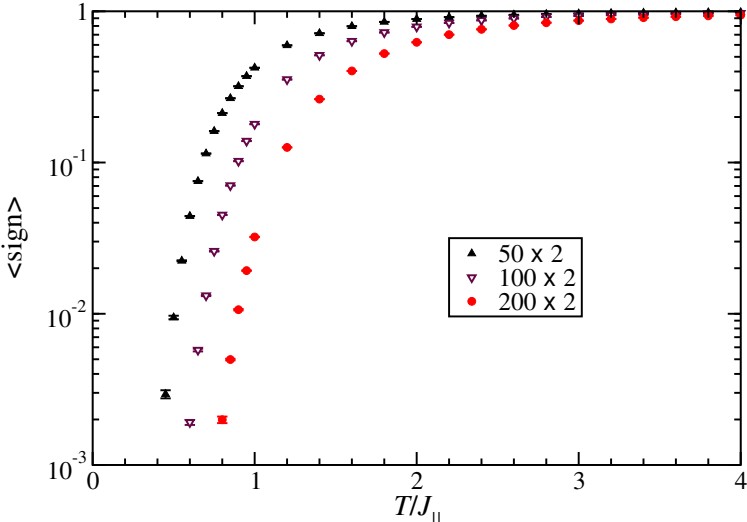

Figure 2: Average sign, ⟨sign⟩, for a QMC simulation in the single-site basis of the frustrated ladder at $J_\perp = 0.38$, $J_\parallel = 1$, and $J_\times = 0.196$, shown as a function of temperature for ladders with $L = 50$, 100, and 200 rungs. Analogous results for the rung basis are shown in Fig. 4 and the thermodynamic response, also computed in the rung basis, in Fig. 12.

|  | $T_i$ | $T_i^z$ | $T_i^+$ | $T_i^-$ | $D_i^z$ | $D_i^+$ | $D_i^-$ |
|---|---|---|---|---|---|---|---|
| $\lvert S\rangle_i$ | 0 | 0 | 0 | 0 | $\lvert 0\rangle_i$ | $-\sqrt{2}\lvert +\rangle_i$ | $\sqrt{2}\lvert -\rangle_i$ |
| $\lvert 0\rangle_i$ | 1 | 0 | $\sqrt{2}\lvert +\rangle_i$ | $\sqrt{2}\lvert -\rangle_i$ | $\lvert S\rangle_i$ | 0 | 0 |
| $\lvert +\rangle_i$ | 1 | 1 | 0 | $\sqrt{2}\lvert 0\rangle_i$ | 0 | 0 | $-\sqrt{2}\lvert S\rangle_i$ |
| $\lvert -\rangle_i$ | 1 | $-1$ | $\sqrt{2}\lvert 0\rangle_i$ | 0 | 0 | $\sqrt{2}\lvert S\rangle_i$ | 0 |

Table 1: Action of local total-spin and spin-difference operators (2) on the local spin-dimer basis states (6). Because $\vec{T}_i^2$ and $T_i^z$ are diagonal in this basis, we quote only the corresponding quantum numbers.

## 3.2 Rung basis

### 3.2.1 Fully frustrated ladder

At the fully frustrated point, $J_\times = J_\parallel$, it is clear from the vanishing of the $\vec{D} \cdot \vec{D}$ term in Eq. (4) that the sign problem is eliminated completely [Eq. (5)] [33, 34]. We choose to sample the partition function in this basis using the stochastic series expansion (SSE) representation with generalized directed loop updates [69, 70]. However, the fact that the Hilbert space is split into sectors characterized by the different local quantum numbers $\{T_i\}$ gives rise to problems with the ergodicity of the conventional SSE algorithm at low temperatures, and these require a parallel-tempering protocol [71–73] to overcome. Further details of our sampling strategy for the fully frustrated ladder can be found in Ref. [33]. The magnetic susceptibility, $\chi(T)$, and specific heat, $C(T)$, are estimated in the usual way for the SSE technique, namely from the fluctuations of the total magnetization and of the expansion order [74], respectively.

### 3.2.2 Partially frustrated ladder

For the partially frustrated ladder, the fact that $J_\times \neq J_\parallel$ mandates working with the more general Hamiltonian of Eq. (4), where the sign problem returns even in the rung basis. We

consider the local basis states on rung $i$,

$$|S\rangle_i = \tfrac{1}{\sqrt{2}}(|\uparrow\downarrow\rangle_i - |\downarrow\uparrow\rangle_i),$$

$$|0\rangle_i = \tfrac{1}{\sqrt{2}}(|\uparrow\downarrow\rangle_i + |\downarrow\uparrow\rangle_i), \qquad |+\rangle_i = |\uparrow\uparrow\rangle_i, \qquad |-\rangle_i = |\downarrow\downarrow\rangle_i, \qquad (6)$$

in terms of which the matrix elements of the operators (2) are given in Table 1. In the SSE framework, the specific requirement is that the matrix elements of $-H$ should be non-negative [69]. This can always be ensured for diagonal matrix elements by addition of a suitable global constant to the Hamiltonian, which only shifts the zero of energy and otherwise has no effect on the physics. If the rewritten Hamiltonian (4) is defined on a bipartite lattice, as is the case for the ladder, one may further perform a $\pi$-rotation around the $z$-axis on one of the two sublattices,

$$T_{2i}^\pm \to -T_{2i}^\pm, \qquad D_{2i}^\pm \to -D_{2i}^\pm, \qquad (7)$$

which preserves the commutation relations (3) but renders the sign of the $T_i^\pm T_{i+1}^\mp$ terms negative, as required.[1]

The prefactor specifying the sign of the $\vec{D} \cdot \vec{D}$ interaction in Eq. (4) is positive for the case $J_\times < J_\parallel$ considered here. Although the $D^z D^z$ terms are off-diagonal and non-negative in the basis (6), as Table 1 makes clear, they pose no problem here [33, 34]. One way to see this explicitly is to send $\vec{D}_{2i} \to -\vec{D}_{2i}$ for one sublattice, which corresponds to an interchange $1 \leftrightarrow 2$ of the two legs on every second rung in the transformation (2). This changes the sign of the $D_i^z D_{i+1}^z$ terms such that they can also be considered to be negative.

These considerations leave only the $D_i^\pm D_{i+1}^\mp$ terms. Inspection of Table 1 shows that their matrix elements can be both positive and negative, such that they do actually give rise to a sign problem. However, the $D_i^\pm D_{i+1}^\mp$ terms also exchange a pair of local quantum numbers, $(T_i, T_{i+1})$. Because such an exchange must be compensated by further similar terms in the SSE operator string, the occurrence of this type of term is severely restricted. The resulting sign problem therefore turns out to be remarkably mild, as we will demonstrate in our numerical results.

We comment finally that the absence of local conservation laws improves the ergodicity of the Monte Carlo sampling. Thus we found that it is not necessary to employ parallel tempering for simulations with any parameter sets where $J_\times \neq J_\parallel$.

## 3.3 Numerical results for the average sign

Because the sign problem is usually worst at low temperatures [24], we have performed a scan to compute the average sign, $\langle\text{sign}\rangle$, throughout the parameter space for a fixed low temperature, $T/J_\parallel = 0.05$, for ladders of $L = 50$ rungs. We note that $T/J_\parallel = 0.05$ is so low that, for the case shown in Fig. 2, it would be impossible to obtain any meaningful results by performing simulations in the single-site basis. By contrast, the results for simulations performed in the rung basis, shown in Fig. 3, are remarkably well-behaved.

In detail, the average sign is indistinguishable from unity over the majority of the phase diagram, and a noticeable deviation appears only within the transition region from the rung-triplet to the rung-singlet phase. Although this deviation is largest in the low-$J_\perp$ region ($J_\perp, J_\times \ll J_\parallel$), we stress that $\langle\text{sign}\rangle > 0.86$ over the entire parameter regime for simulations with $L = 50$ at $T/J_\parallel = 0.05$. Qualitatively, the maximum of the difference $1 - \langle\text{sign}\rangle$ in Fig. 3 traces the phase transition line (Fig. 1).

The behavior of the average sign is investigated more closely for a ladder with $J_\perp/J_\parallel = 0.38$ in Fig. 4. The inset shows QMC data for $\langle\text{sign}\rangle$ as a function of $J_\times/J_\parallel$ at various low temperatures. As noted in Sec. 2, this transition region between the rung-triplet and -singlet phases

---

[1]This sublattice rotation ensures that all of the corresponding configurations always have positive weight and thus need not actually be carried out explicitly; one source for further details is Ref. [1].

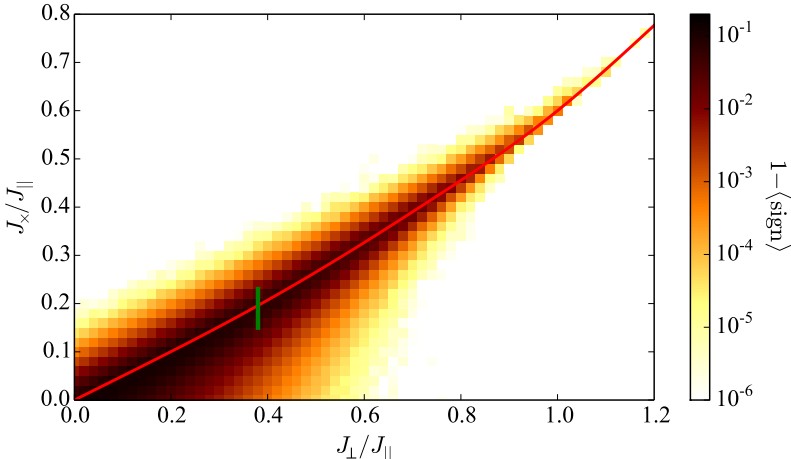

Figure 3: Deviation of the average sign from one, $1 - \langle\text{sign}\rangle$, computed by QMC simulations in the rung basis using ladders of $L = 50$ rungs at $T/J_\parallel = 0.05$. The solid red line is the phase transition from Fig. 1 and the vertical green line shows the scan direction detailed in the inset of Fig. 4. White regions correspond to a deviation in average sign of less than $10^{-6}$; outside the range of parameters covered by the figure, $\langle\text{sign}\rangle$ is numerically indistinguishable from 1.

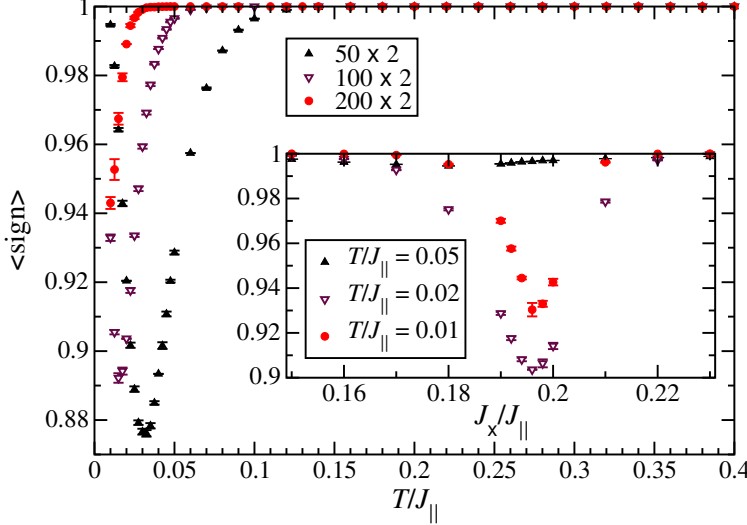

Figure 4: Average sign, $\langle\text{sign}\rangle$, for a QMC simulation in the rung basis using the parameters $J_\perp = 0.38$, $J_\parallel = 1$, and $J_\times = 0.196$, shown as a function of temperature for ladders with $L = 50$, 100, and 200 rungs. The thermodynamic response of this system is shown in Fig. 12. Inset: $\langle\text{sign}\rangle$ shown as a function of $J_\times$ for a ladder with $J_\parallel = 1$, $J_\perp = 0.38$, and $L = 100$ rungs, at three temperatures, $T/J_\parallel = 0.01$, 0.02, and 0.05.

has been studied intensively, in particular with regard to a possible intermediate dimerized phase [38–42]. While a rather sharp dip can be identified in $\langle\text{sign}\rangle$ for $J_\times/J_\parallel \approx 0.2$ in the data for $T/J_\parallel = 0.02$ and 0.01, the absolute value of $\langle\text{sign}\rangle$ remains above 0.9 at all times, meaning that QMC simulations performed in this regime are fully reliable. Surprisingly, the average sign shows a non-monotonic temperature-dependence, increasing at the lowest temperature considered here, $T/J_\parallel = 0.01$, compared to $T/J_\parallel = 0.02$. The position of the minimum in $\langle\text{sign}\rangle$

at sufficiently low temperatures appears to constitute a signature of the phase transition that is comparable in accuracy with dedicated $T = 0$ methods [38–42], and thus we obtain the estimate $J_{\times,c}/J_\parallel \approx 0.196$ at $J_\perp/J_\parallel = 0.38$.

The main panel of Fig. 4 shows $\langle\text{sign}\rangle$ as a function of temperature for three different system sizes at the estimated critical value, $J_{\times,c}/J_\parallel \approx 0.196$ for $J_\perp/J_\parallel = 0.38$. In contrast to the single-site basis (Fig. 2), we observe deviations of the average sign from unity only for temperatures $T/J_\parallel < 0.1$. Our results display not only the aforementioned non-monotonic temperature-dependence, whereby $\langle\text{sign}\rangle$ increases towards lower temperatures, but also that, quite unlike Fig. 2, the average sign increases with increasing system size.

The fact that the average sign is close to unity may be expected deep inside the rung-singlet and -triplet phases, where the respective ground states are thought to be well captured by the rung basis. Only on approaching the phase transition, and indeed only at rather weak $J_\perp/J_\parallel$, do we observe appreciable deviations from unity, indicating that inter-rung fluctuations are large. Still, with values $\langle\text{sign}\rangle \gtrsim 0.9$, the average sign remains remarkably well-behaved throughout the phase diagram. In particular, the fact that its behavior improves with decreasing temperature and with increasing system size is in sharp contrast to expectations [24] and to the situation in the single-site basis (Fig. 2), and offers still greater potential for minimizing the sign problem.

To account for this behavior at a qualitative level, in Subsec. 3.2.2 we pointed out that configurations with a negative sign are severely constrained for the frustrated ladder. Indeed, our empirical observation is that the sign problem in the rung basis is completely absent for systems with open boundary conditions, while for periodic boundary conditions the first term giving rise to a sign problem occurs at order $L + 1$ and contains an operator string that wraps around the entire system, as also noted in Ref. [34]. This is consistent with the fact that a significant number of configurations with negative sign arises only in a small temperature window, as well as with the fact that such configurations never constitute a macroscopic fraction of all configurations.

Despite the complete absence of a sign problem when working in the rung basis on a ladder with open boundaries, periodic boundary conditions offer several advantages in the calculation of bulk thermodynamic properties for finite systems. Not least of these is the elimination of surface terms, which could result in some particular problems in the Haldane phase of the ladder. In any case, when $\langle\text{sign}\rangle \gtrsim 0.9$ it is no longer true that the sign "problem" is a limiting factor for QMC simulations, because other algorithmic aspects, such as autocorrelation times, become more relevant for simulation efficiency. Specifically, if $\langle\text{sign}\rangle \approx 0.9$ then it is necessary to collect only 25% more samples in order to compensate for the loss of accuracy associated with sign effects. We remark in closing that the single-site basis remains nevertheless the natural choice for QMC simulations of unfrustrated ladders [11–15, 33], and accordingly we do not pursue the case $J_\times = 0$ here.

## 4 Results: Thermodynamic properties

In Figs. 5–12 we present numerical results for the magnetic specific heat and susceptibility of frustrated spin ladders for a selection of representative points in the phase diagram, as shown by the corresponding numbered circles in Fig. 1. In these figures, the sizes of the QMC error bars are in general significantly smaller than the symbol sizes, demonstrating the high quality of the nearly-sign-free simulations. For interpretive purposes we include certain analytical results where appropriate and for numerical comparison we also include exact diagonalization

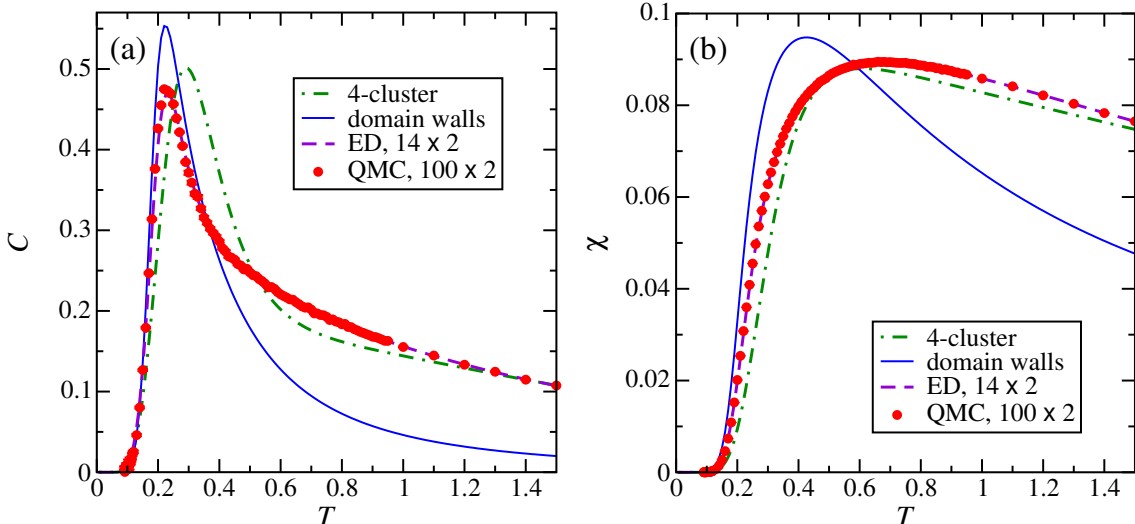

Figure 5: Magnetic specific heat, $C$ (left panel), and susceptibility, $\chi$ (right panel), shown per spin for a fully frustrated ladder with rung coupling $J_\perp = 1.45$ and inter-rung couplings $J_\times = J_\parallel = 1$. We compare QMC results obtained for a ladder of $L = 100$ rungs with ED results for $L = 14$ and with approximate calculations based on clusters of 4 rungs and on a model of non-interacting domain walls.

(ED) results obtained for small systems.[2] At points sufficiently deep inside the rung-singlet phase, finite-size effects are negligible at all temperatures. In these cases, and at all high temperatures, the ED results coincide exactly with the QMC ones, validating again the reliability of the simulations.

## 4.1 Fully frustrated ladder

We begin by presenting for reference two examples of the thermodynamic response of fully frustrated ladders ($J_\times = J_\parallel$), a case already investigated in some detail in Ref. [33]. We choose first a point in the rung-singlet phase but close to the phase transition, $J_\perp / J_\parallel = 1.45$, and present numerical results for the specific heat [Fig. 5(a)] and susceptibility [Fig. 5(b)]. On a technical level, it is important to note that the ED results for $L = 14$ and QMC for $L = 100$ rungs are in good agreement for both quantities, with discrepancies appearing only in the specific heat and only around its maximum. Thus we conclude that the QMC results for $L = 100$ rungs in Fig. 5 are indistinguishable from the thermodynamic limit (indeed, the ED data for $L = 14$ already yield a good approximation to the infinite system).

$\chi(T)$ in Fig. 5(b) exhibits a broad maximum, which is characteristic for highly frustrated magnets. Its temperature, $T^\chi_{\max} \simeq 0.68 J_\parallel$, is not particularly low in comparison to the one-triplon energy, $\tilde{E}_{n=1} = J_\perp = 1.45 J_\parallel$. However, as discussed in Ref. [33], a characteristic quantity more useful than the broad maximum is that $\chi$ attains half of its maximal value at the comparatively low temperature $T^\chi_{\text{half}} \simeq 0.248 J_\parallel$. The rise of the specific heat in Fig. 5(a) occurs at an even lower temperature, $T^C_{\text{half}} \simeq 0.169 J_\parallel$, in accord with the fact that $C(T)$ is sensitive to the lowest singlet bound state, which appears at $\tilde{E}^1_{n=2} = 0.9 J_\parallel$ for these parameters. However, the most striking feature of Fig. 5(a) is clearly the emergence of a remarkably sharp maximum

---

[2]ED results are based on the Hamiltonian of Eq. (1) whenever $J_\times \neq J_\parallel$. With the help of the spatial symmetries and conservation of total $S^z$, it is straightforward to perform full diagonalization of a system with up to 20 spins 1/2 ($L \leq 10$ rungs). At $J_\times = J_\parallel$, represented here by Figs. 5 and 6, one may take advantage of the additional local conservation laws to further simplify the problem [33,75–77]. Although the complexity remains exponential in $L$, the rewriting (5) makes it possible to access $L = 14$ rungs (28 $S = 1/2$ spins) [33].

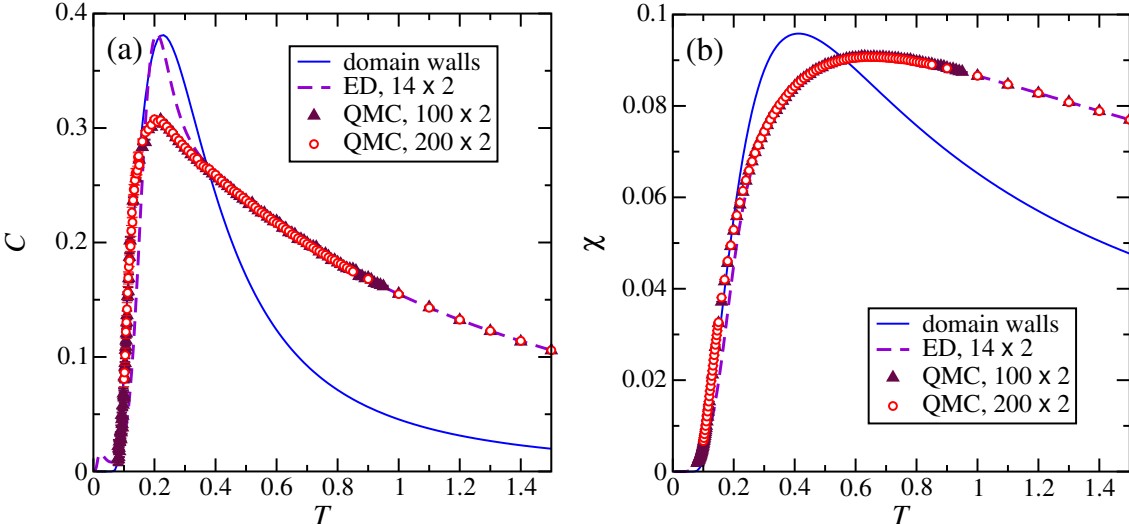

Figure 6: Magnetic specific heat, $C$, and susceptibility, $\chi$, shown per spin for a fully frustrated ladder with rung coupling $J_\perp = 1.4$ and inter-rung couplings $J_\times = J_\parallel = 1$. These values lie very close to the critical line in Fig. 1. We compare QMC results obtained for ladders of $L = 100$ and 200 rungs with ED results for $L = 14$ and with a model of non-interacting domain walls.

in $C(T)$ at the very low temperature $T_{\max}^C \simeq 0.231 J_\parallel$.

These results were interpreted in Ref. [33] by a detailed analysis of the many rung-triplet bound states in the rung-singlet phase, which move to anomalously low energies near the quantum phase transition. For illustration, we appeal to two analytical approximations that yield the other two curves included in each panel of Fig. 5. First, a computation taking into account all excited states of $n$-triplon clusters, with $n \leq 4$, yields an accurate description deep in the rung-singlet phase [33] and also a good account of the high-temperature behavior at a point as close to the transition as $J_\perp/J_\parallel = 1.45$. However, it cannot provide an accurate reproduction of either the sharp nature or the low effective temperature scale of the specific-heat peak. For this, a "domain-wall" model [33] yields a better description, particularly of the low-temperature onset ($T_{\mathrm{half}}^C$) and maximum position ($T_{\max}^C$) in $C(T)$. Near the transition, these walls exist between domains of almost-degenerate rung-singlet and -triplet states, and describe the contributions of $S = 1/2$ end-spins terminating the $n$-site rung-triplet (spin-1) chain segments in a rung-singlet background for all values of $n$ [78–80]. We conclude that the characteristic emergent temperature scales ($T_{\mathrm{half}}^C$, $T_{\max}^C$, and $T_{\mathrm{half}}^\chi$) reflect not only the energies of the low-lying excited states but also their degeneracies, and can therefore be considered as signatures of the complex spectrum of bound states.

Figure 6 presents thermodynamic results for the point $J_\perp/J_\parallel = 1.4$, which lies effectively at the phase transition for $J_\times = J_\parallel$ (Fig. 1). We observe that the QMC data for $L = 100$ and 200 rungs are very close, showing that $L = 100$ can be considered as fully representative of the thermodynamic limit, even at the phase transition itself. Nonetheless, finite-size effects visible in the ED results are significantly stronger than in Fig. 5. In particular, the $L = 14$ curve in Fig. 6(a) suggests a sharp low-temperature maximum in $C(T)$, whereas the QMC results demonstrate that in fact only a shoulder survives in the thermodynamic limit at the critical coupling (as a consequence of contributions from $n$-rung triplet clusters of all length scales [33]).

The characteristic temperatures observed at $J_\perp/J_\parallel = 1.45$ are further reduced at $J_\perp/J_\parallel = 1.4$, with (from the QMC data in Fig. 6) $T_{\max}^C \simeq 0.199 J_\parallel$, $T_{\mathrm{half}}^C \simeq 0.112 J_\parallel$, and $T_{\mathrm{half}}^\chi \simeq 0.179 J_\parallel$, reflecting a further reduction of the relevant excitation energies [33]. While

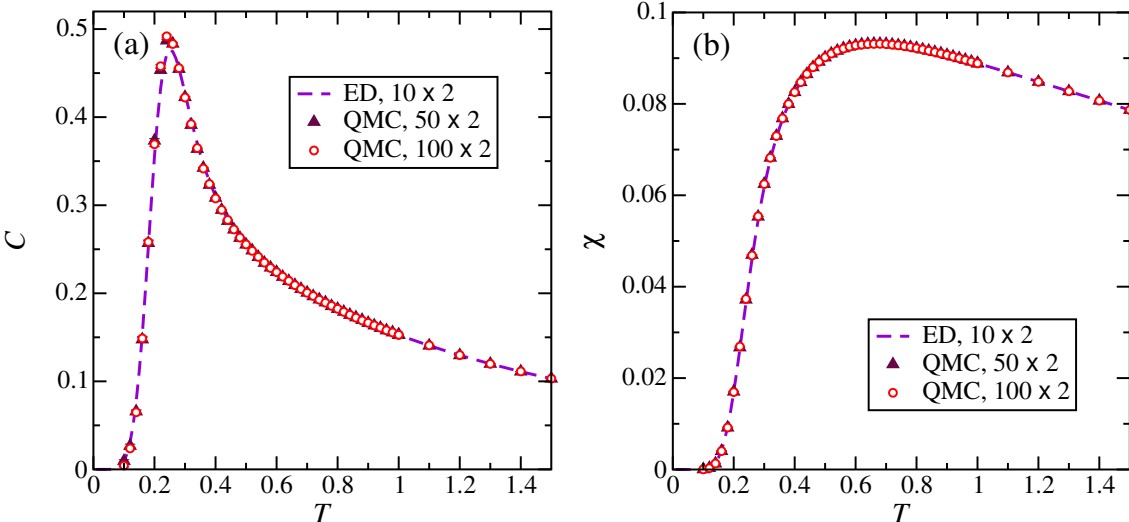

Figure 7: Magnetic specific heat, $C$, and susceptibility, $\chi$, shown per spin for a partially frustrated ladder with rung coupling $J_\perp = 1.4$, leg coupling $J_\parallel = 1$, and diagonal coupling $J_\times = 0.9$. We compare QMC results obtained for ladders of $L = 50$ and 100 rungs with ED results for $L = 10$.

the short-cluster approximation is not well suited to this case, the domain-wall model continues to yield an accurate description at least of the low-temperature onset of both $C(T)$ and $\chi(T)$, demonstrating the importance of these walls as the effective lowest-energy excitations.

## 4.2 Highly frustrated ladder

We extend our considerations by exploring the broader regions of the phase diagram (Fig. 1). To assess the influence of the local conservation laws effective at $J_\times = J_\parallel$, we first move only a little away from full frustration. We consider the illustrative case $J_\perp/J_\parallel = 1.4$, $J_\times/J_\parallel = 0.9$, which lies in the rung-singlet phase but remains close to the transition. Figure 7 presents numerical results for $C(T)$ and $\chi(T)$ of this still highly frustrated ladder. The QMC procedure is now based on the expression of Eq. (4), and while a sign problem is in general present when $J_\times \neq J_\parallel$, the average sign remains numerically indistinguishable from 1 (Sec. 3.3). In Fig. 7 we include ED results for $L = 10$ rungs, which exhibit only minor finite-size effects around the maximum of $C(T)$ at $T/J_\parallel \approx 0.25$. The QMC data for $L = 50$ can definitely be regarded as representative of the thermodynamic limit.

The parameters of Fig. 7 were chosen to give a point approximately the same distance from the phase-transition line as that shown for a fully frustrated ladder in Fig. 5. The overall qualitative features are very similar, including in particular the low onset temperatures and the low-temperature maximum in $C(T)$. This result demonstrates that the qualitative features of the spectrum of multi-triplet bound states [33], and its evolution on approaching the transition, are not strongly affected by the imperfect frustration. Thus while the fully frustrated line and the presence of local conservation laws are helpful for an analytical understanding, including of the behavior at all finite temperatures, they do not lead to any unique physical properties not present in the more general frustrated ladder.

Figure 8 presents the example $J_\perp/J_\parallel = 1.32$, $J_\times/J_\parallel = 0.9$, which is again close to full frustration and lies essentially on the first-order transition between the rung-singlet and -triplet phases (Fig. 1). The thermodynamic response, including the low onset temperatures, a shoulder-type feature in the specific heat at low temperatures, and even the extent of finite-size effects, is very similar to the fully frustrated case shown in Fig. 6. This reinforces the obser-

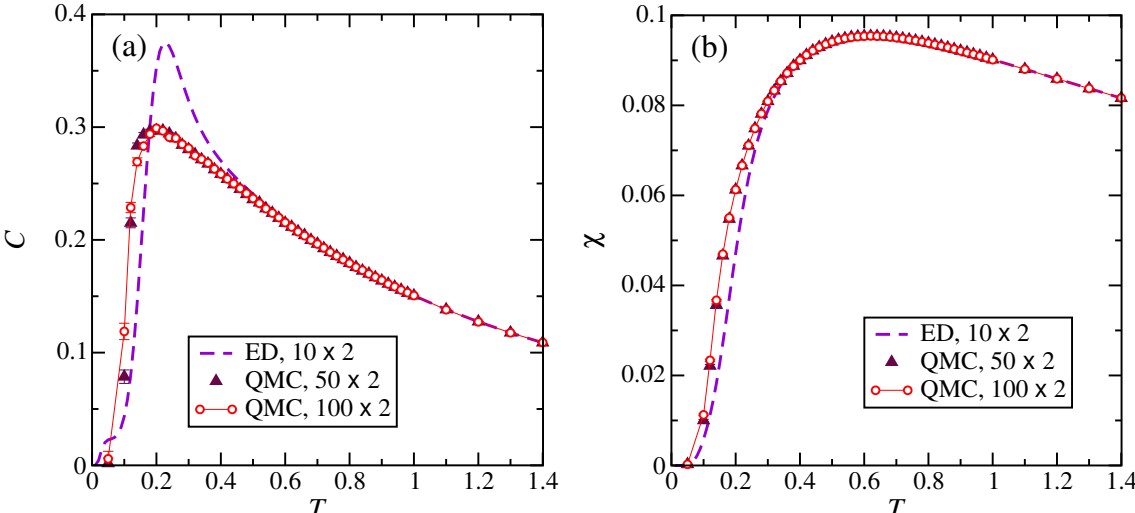

Figure 8: Magnetic specific heat, $C$, and susceptibility, $\chi$, shown per spin for a ladder with rung coupling $J_\perp = 1.32$, leg coupling $J_\parallel = 1$, and diagonal coupling $J_\times = 0.9$. We compare QMC results obtained for ladders of $L = 50$ and 100 rungs with ED results for $L = 10$. The line connecting the $L = 100$ QMC data points is a guide to the eye.

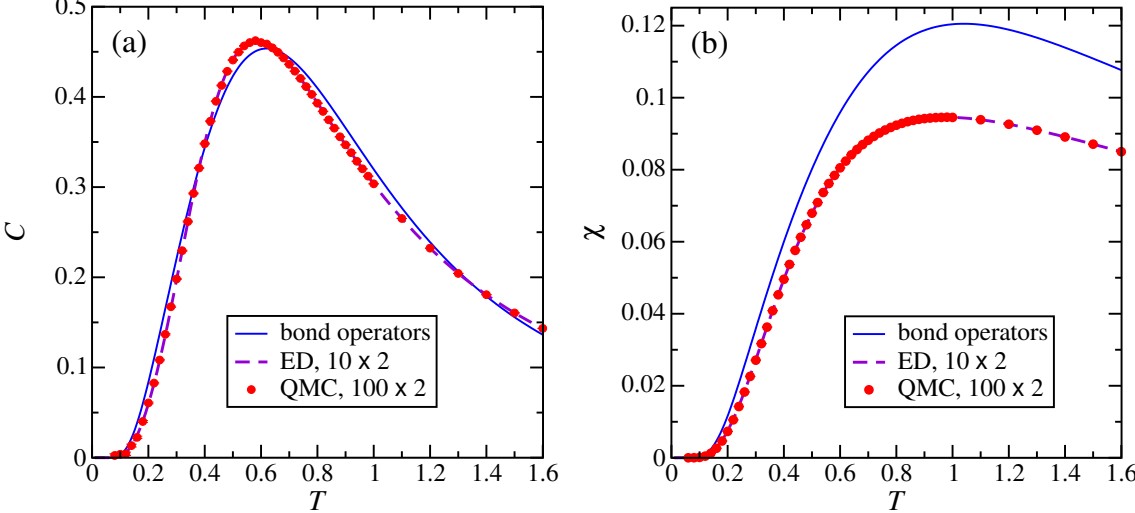

Figure 9: Magnetic specific heat, $C$, and susceptibility, $\chi$, shown per spin for a partially frustrated ladder with rung coupling $J_\perp = 1.45$, leg coupling $J_\parallel = 1$, and diagonal coupling $J_\times = 0.2$. We compare QMC results obtained for ladders of $L = 100$ rungs with ED results for $L = 10$ and the bond-operator result for a corresponding unfrustrated ladder with $J_\perp = 1.45$ and effective leg coupling $J_{\parallel,\text{eff}} = 0.8$.

vation that the local conservation laws present at $J_\times = J_\parallel$ are useful for interpretation but are not essential in determining the qualitative thermodynamic features of the frustrated ladder, at least in the regime of small "detuning" $J_\times \neq J_\parallel$.

## 4.3 Rung-singlet phase

We turn next to the behavior of the frustrated ladder far from the phase transition. Remaining in the rung-singlet phase, Fig. 9 presents results for a point deep inside this regime, $J_\perp/J_\parallel = 1.45$, $J_\times/J_\parallel = 0.2$. Here we find that the data for $L = 10$ and 100 rungs are indistin-

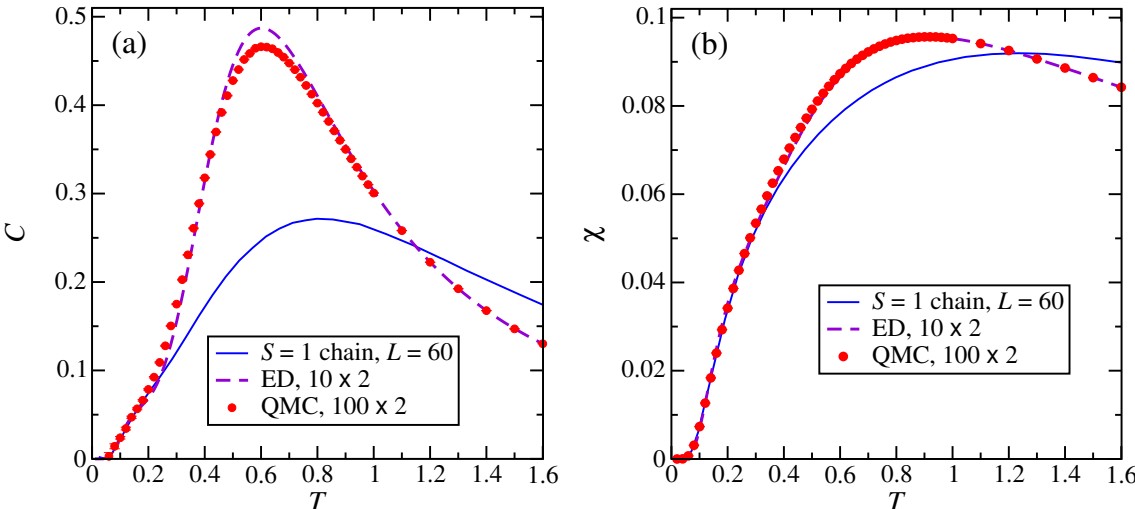

Figure 10: Magnetic specific heat, $C$, and susceptibility, $\chi$, shown per spin for a partially frustrated ladder with rung coupling $J_\perp = 0.38$, leg coupling $J_\parallel = 1$, and diagonal coupling $J_\times = 0.9$. We compare QMC results obtained for ladders of $L = 100$ rungs with ED results for $L = 10$, and with QMC results for a spin-1 chain of $L = 60$ sites [22] with $J_{\text{eff}} = 0.95$, which are normalized to twice the number of spins in the chain.

guishable, signaling a system with a short correlation length. For an analytical understanding of this case, we consider a bond-operator treatment [81] of the frustrated ladder. In this approach, $J_\times$ appears on exactly the same footing as $J_\parallel$, giving an effective unfrustrated ladder with leg coupling $J_\parallel - J_\times$, which is expected to provide a good description of the thermodynamic response far from full frustration. In Fig. 9, we illustrate this by comparing our QMC data with bond-operator results for a ladder with rung coupling $J_\perp = 1.45$ and leg coupling $J_{\parallel,\text{eff}} = 0.8$, where the gap, $\Delta = 0.575 J_\perp$, corresponds to a correlation length $\xi \propto 1/\Delta$ of order 3 lattice constants [13]. The effective unfrustrated model gives an excellent description of $C(T)$ for the weakly frustrated ladder, indicating that all the complexity of low-lying bound states is absent in this regime. However, we observe that the effective model is not able to reproduce in full the flattening of the peak in $\chi(T)$ caused by the presence of frustration. We comment that there is no longer any particularly low temperature scale characterizing the response of the system for these parameters.

## 4.4 Rung-triplet phase

Figure 10 presents the contrasting results for a point deep within the rung-triplet phase, $J_\perp/J_\parallel = 0.38$, $J_\times/J_\parallel = 0.9$. From Eq. (4), the low-energy physics of this case should be similar to an $S = 1$ Heisenberg chain with an effective coupling constant $J_{\text{eff}} = (J_\parallel + J_\times)/2 = 0.95 J_\parallel$. We note first that finite-size effects in the specific heat [Fig. 10(a)] are comparable to those observed in the fully frustrated ladder at $J_\times = J_\parallel = J_\perp = 1$ [33]. These are to be expected because the ED system size, $L = 10$, is not significantly larger than the correlation length, $\xi \approx 6$, of the spin-1 chain at $T = 0$ [56, 57]. To test this comparison in full, Fig. 10 includes properly rescaled QMC results for a spin-1 Heisenberg chain with $L = 60$ sites [22]. It is clear that these do match the low-temperature asymptotics of the frustrated ladder, and from them we identify the Haldane gap [56, 57], $\Delta \approx 0.4105 J_{\text{eff}} \approx 0.39 J_\parallel$, as the lowest energy and temperature scale for the parameters of Fig. 10. At higher temperatures, we observe strong additional contributions beyond the effective spin-1 chain, in particular in $C(T)$, and these can be attributed to the thermal population of mostly localized rung-singlet excitations, as

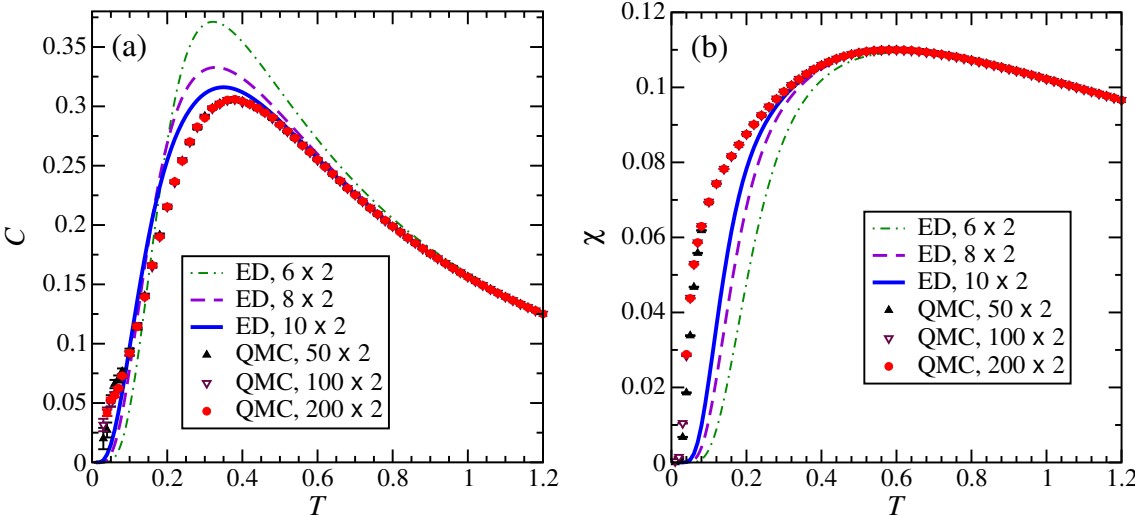

Figure 11: Magnetic specific heat, $C$, and susceptibility, $\chi$, shown per spin for a ladder with rung coupling $J_\perp = 1$, leg coupling $J_\parallel = 1$, and diagonal coupling $J_\times = 0.599$. We compare QMC results obtained for ladders of $L = 50$, 100, and 200 rungs with ED results for $L = 6$, 8, and 10.

discussed for the fully frustrated ladder in Ref. [33].

## 4.5 Phase transition and weakly coupled chains

We conclude our survey of the phase diagram by returning to its most complex region, the phase-transition line. Here we will show that it remains possible to obtain highly accurate results by QMC in the rung basis, whereas ED is affected by severe finite-size effects at low temperatures.

Figure 11 shows thermodynamic results for the point $J_\perp/J_\parallel = 1$, $J_\times/J_\parallel = 0.599$, which is located on the transition line, but moved from Figs. 6 and 8 in the direction of weakly coupled chains. Once again, all QMC results with $L \geq 50$ rungs are consistent and thus can be considered as representing the thermodynamic limit. However, the finite-size effects visible in the ED results are distinctly enhanced, by comparison with Fig. 8, at all temperatures below the peaks in $C(T)$ and $\chi(T)$. A $T = 0$ density-matrix renormalization-group (DMRG) investigation [37] found a gap $\Delta/J_\parallel \approx 0.13$ for the parameters of Fig. 11. Despite the appearance of anomalous features such as the shoulder in the low-temperature specific heat, visible in Fig. 11(a), our results are indeed consistent with a finite gap, albeit one with the nature of an emergent low energy scale. As a consequence, we conclude that the $T = 0$ transition from the rung-singlet to the rung-triplet phase remains of first-order type for $J_\times/J_\parallel \geq 0.6$.

Moving yet further in the direction of weak interchain coupling, Fig. 12 shows results for a ladder with $J_\perp/J_\parallel = 0.38$ and $J_\times/J_\parallel = 0.196$. The average sign appearing in the QMC simulations for this system was shown in Fig. 4 and we observe that its deviations from unity remain sufficiently small that they do not impact the accuracy of the QMC results. However, the finite-size effects in the ED calculations are now enhanced very dramatically, and even in the QMC simulations it is clear that the $L = 200$ data are required to ensure a good approximation to the thermodynamic limit for the features of $C$ and $\chi$ at the lowest temperatures; in particular, the susceptibility is inordinately sensitive at $T/J_\parallel < 0.05$ [Fig. 12(b)]. From our observation (Sec. 3.3) that the sign problem arises due to a boundary term, it is naturally sensitive to the correlation length of the system and thus its most serious manifestations (Figs. 3 and 4), as well as the most serious finite-size effects (Fig. 12), occur when the correlation length is

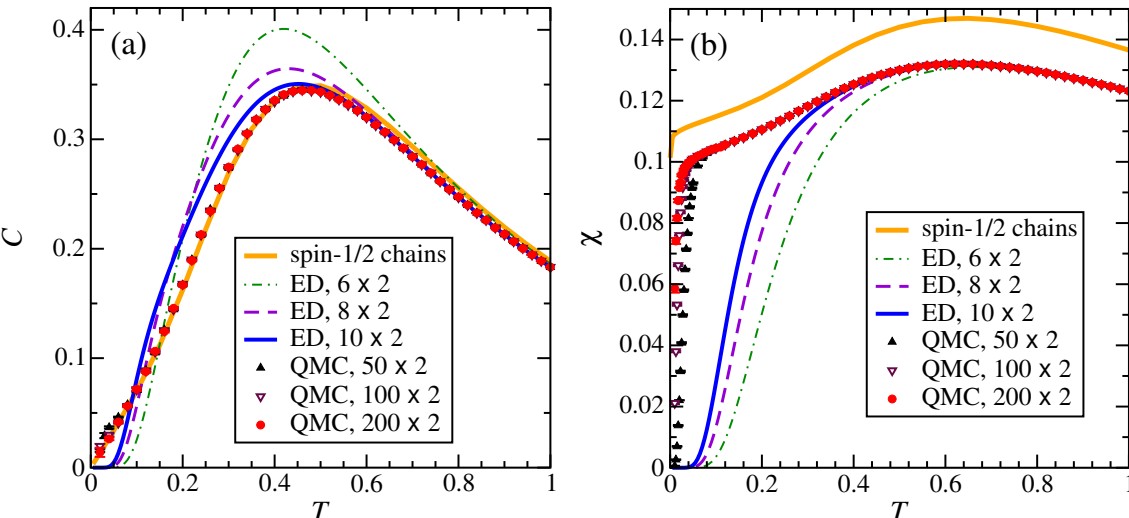

Figure 12: Magnetic specific heat, $C$, and susceptibility, $\chi$, shown per spin for a ladder with rung coupling $J_\perp = 0.38$, leg coupling $J_\parallel = 1$, and diagonal coupling $J_\times = 0.196$. We compare QMC results obtained for ladders of $L = 50$, 100, and 200 rungs with ED results for $L = 6, 8$, and 10, as well as with exact results for two decoupled spin-1/2 chains, with coupling constant $J_\parallel = 1$, in the thermodynamic limit [82].

largest, explaining directly why $1 - \langle \text{sign} \rangle$ at fixed temperature acts as an excellent indicator of the phase transition in parameter space.

To interpret the results in Fig. 12, we show the exact specific heat [82] and susceptibility [82, 83] of two isolated spin-1/2 chains. Our numerical data for $C(T)$ converge to a curve whose low-temperature behavior is captured remarkably well by a pair of decoupled chains, implying that for this parameter regime the rung and diagonal interactions act to cancel each other. The shape of $\chi(T)$ for this ladder also follows very closely the result for two decoupled chains, where it is known to approach a logarithmic singularity at low $T$, although the absolute scale is renormalized by a factor of approximately 0.9. We expect that this deviation reflects in part the sensitivity of the $S = 1/2$ degrees of freedom in the ladder legs to the confining effects of the relevant rung-coupling perturbation and in part the sensitivity to this coupling of matrix-element effects. Thus our results quantify the extent to which the ladder legs may be regarded as effectively decoupled in the presence of finite but mutually antagonistic rung and diagonal interactions. An essential qualitative observation is that our numerical data for $C$ and $\chi$ are most easily reconciled with a direct and continuous transition at $T = 0$ between the rung-singlet and rung-triplet phases at $J_\perp = 0.38$, with no evidence for the presence of an intermediate phase.

### 4.6 Frustrated ladder physics

We conclude our discussion of the physics of the frustrated two-leg ladder with a brief summary. With the possible exception of a vanishingly narrow region at small interchain coupling, the frustrated ladder has only two ground states, a rung-singlet phase when the rung coupling significantly exceeds the leg and diagonal couplings and a rung-triplet phase in the opposite situation. Far from the quantum phase transition separating the two states, the rung-singlet phase can be described rather well by an effective unfrustrated ladder model (Fig. 9) while the rung-triplet phase contains the physics of both quasi-localized bound states and the extended states of an effective spin-1 chain (Fig. 10).

In the strongly coupled and strongly frustrated ladder, the transition is strongly first-order.

This regime is characterized by large numbers of low-lying bound states, which manifest themselves in the sharp peak in $C(T)$ and very low onset temperature in $\chi(T)$ (Figs. 5 and 7). However, exactly at the transition, the diverging number of these states causes the sharp specific-heat peak to vanish (Figs. 6 and 8). It is clear that the behavior of the manifold of low-lying bound states around the transition is not very sensitive to whether the ladder is perfectly frustrated (Figs. 6 and 8) or not (Figs. 5 and 7).

As the phase-transition line is followed to weaker interchain couplings, there is a clear change of behavior, consistent with a closing of the gaps of both the rung-singlet and -triplet phases (Fig. 11). Previous DMRG and field-theoretical studies indeed suggest a gapless state and continuous transitions across the line in this regime. While our QMC results cannot exclude the possibility that this topological transition remains of first order with an exponentially small gap, it is clear that the physics of the frustrated ladder with weakly coupled chains (Fig. 12) is very different from the strongly coupled regime (Figs. 5–8). We suggest that DMRG may be the most appropriate technique for revisiting the nature of the quantum phase transition on the approach to two decoupled chains, for example by computing the central charge and the evolution of the gaps in both phases.

## 5 Summary and perspectives

We have demonstrated that the sign problem which plagues QMC studies of frustrated quantum spin models can be rendered so weak as to be irrelevant in certain classes of system. Taking the example of the frustrated two-leg spin-1/2 ladder, we have shown that efficient QMC simulations can be performed throughout the entire phase diagram, and that ladders of size 200 rungs access the thermodynamic limit in all cases. The key to the success of our improved QMC approach is to rewrite the Hamiltonian in a different basis, which can be regarded as forming a composite-spin system out of the original single-site basis; in the ladder, the natural choice is a basis of ladder rungs.

Complete elimination of the sign problem is achieved whenever a rewriting of the form of Eq. (5) is possible, which leads to a composite-spin model whose basis units themselves form an unfrustrated lattice [33, 34]. This situation arises not only in the fully frustrated case of the present ladder model but also in a considerable number of highly frustrated spin models whose frustration results in local conservation laws [44, 45, 75–77, 80, 84–99] for clusters forming a bipartite lattice. While a square-lattice bilayer analog of the fully frustrated ladder has recently been studied by QMC simulations [34, 35], further cases in the same class remain to be investigated. Although the example of the spin-1/2 $J_1$–$J_2$ chain [25, 26] and some of our own results for ladders far from perfect frustration demonstrate that a rewriting of the Hamiltonian is not a prerequisite for performing QMC, it remains true that appropriately chosen cluster bases provide a general tool for optimizing the efficiency of QMC simulations.

The fact that the sign problem remains mild in the entire space of parameters for the ladder with $J_\times \neq J_\parallel$ can be traced to the fact that only $\vec{D} \cdot \vec{D}$ terms appear in the rung-basis Hamiltonian of Eq. (4). We believe that such favorable behavior is a generic feature of models where a global exchange symmetry of the two ends of each dimer ensures the absence of $\vec{T} \cdot \vec{D}$ terms. This includes frustrated magnets in higher dimensions, where few accurate numerical methods exist. However, $\vec{T} \cdot \vec{D}$ terms do appear in more general models when expressed in a dimer basis, not least the asymmetrically frustrated ladder, and the Shastry-Sutherland model [100, 101] provides a key example in two dimensions. We defer to future studies a detailed investigation of the extent to which suitably chosen cluster bases render QMC simulations feasible for higher-dimensional models. Here we comment only that the search for such an optimal cluster basis in any given frustrated spin system may also be extended by performing a systematic scan of

the manifold of basis transformations [68].

We remark also that the sign problem remaining for the frustrated ladder in the rung basis takes a new and quite unconventional form, in which it becomes weaker as a function of increasing system size and is maximal at a finite temperature before the average sign recovers to unity at low $T$. From a practical standpoint, these results offer the possibility of working around the sign problem for any given parameter set. From an analytical point of view, they provide new insight into the nature and number of the negative-weight configurations, which in the frustrated ladder model appear to be a boundary problem only (and hence a set of vanishing measure in the thermodynamic limit). Finally, once QMC simulations have been demonstrated to work efficiently for the computation of static thermodynamic properties, it will be of great value to apply them also to the computation of dynamical response functions at finite temperatures.

# Acknowledgements

We thank F. Alet, K. Damle, R. Noack, G. Radtke, and O. Vaccarelli for helpful discussions.

**Funding information:** This work was supported by the DFG research unit FOR1807 and by the Swiss NSF.

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
