# Peer review of "Efficient Quantum Monte Carlo simulations of highly frustrated magnets: the frustrated spin-1/2 ladder"

_SciPost Physics, doi:SciPost Phys. 3, 005 (2017)_

## Round 1 · Referee Report · Anonymous · 2017-5-9

Strengths
This is a very extended study of strongly frustrated antiferromagnetic spin ladder problem. It represents tour de force quantum Monte Carlo investigation of the important model problem in frustrated magnetism. The key technical advance that allowed the authors to obtain their remarkable results consists in successfully choosing a sign-problem "free" basis, the dimer basis. The elimination of the sign-problem is not complete, as the authors explain, but is dramatic nonetheless. It allows for an almost complete numerical solution of the problem. This is an important advance which should be shared with the community. The fact that only two phases are present in the wide range of interchain interactions is now firmly established.
Quite importantly, key technical steps and arguments are very well explained, which makes the manuscript accessible to a wide audience well beyond few numerical experts (who, nonetheless, will too find this work very important).
Weaknesses
There is no real weakness in the presented study. The range of small interchain interactions $J_\perp$ and $J_\times$ remains subject to strong sign problem, as Fig.3 shows. This, rather than the weakness, represents an outstanding open problem numerous attempt to analyze which numerically (many of which are discussed in the text) have basically failed so far.
Report
Very solid and exhaustive work which needs to be published. Really high quality study with important findings.
Requested changes
I would like to ask the authors to comment on the range of weak $J_\perp$ and $J_\times$ within which, in authors' opinion, columnar dimerized phase proposed in Ref.59, is possible. It is also quite possible that such a phase is absent from the phase diagram altogether and the transition between rung-triplet and rung-singlet phases is of the first order in the full range of antiferromagnetic $J_\perp$ and $J_\times$. I think it may be interesting for readers to know what authors opinion about this so far unresolved issue is.
Closely related to this issue is the observation, made in Ref.61, that changing the sign of interchain exchanges $J_\perp$ and $J_\times$ to the ferromagnetic one greatly simplifies RG flow of the problem and does produce, according to DMRG results of ref.61, the above mentioned columnar dimer phase. In the context of the current study this seems to imply that negative sign of $J_\perp$ and $J_\times$ should result in even weaker sign problem. This, in the reviewer's opinion, would represent quite an unexpected development. (I would like to add there that this point is only an open question/suggestion and should not be treated as a "requested change". I do not insist on addressing it in the current manuscript.)
Author: Andreas Honecker on 2017-05-31 [id 140]
(in reply to Report 1 on 2017-05-09)
We are very grateful to the referee for his of her appreciation of our work; it is extremely gratifying to read the assessment: "Very solid and exhaustive work which needs to be published. Really high quality study with important findings."
Concerning the referee's "Requested changes":
first, regarding the possible presence of the proposed columnar dimerised phase, in the previous version of the manuscript we stated at the very end of section 2.2 that, on the basis of our results, we believe such a phase to be "invisibly narrow" if it exists at all. However, because our simulations do not offer any insight at a level deeper than that achieved by previous studies, we believe that it is not appropriate to speculate on the existential question. If the absence of this phase requires a first-order transition at all points along the line, the results we report in the final sentence of section 4.5 imply that either the width of the phase or the size of the first-order step is extremely small.
secondly, concerning the RG flow, we certainly have no deeper insight into this than the authors of Ref. [61]. We can state that we find the numerical (DMRG) evidence of Ref. [61], in favour of a robust intermediate phase on changing the sign of certain exchange constants, to be convincing. However, because our results do not improve upon the understanding developed in Ref. [61], that such an intermediate phase remains elusive for the original problem (all exchange constants antiferromagnetic), we prefer to refrain from further, and of necessity speculative, comments in the manuscript.
Author: Andreas Honecker on 2017-06-14 [id 143]
(in reply to Report 3 on 2017-05-29)We are grateful to the referee for her/his careful reading of our manuscript and her/his positive evaluation.
The referee requests one change. However, we did discuss the breaking of leg symmetry, for example in the last but one paragraph of the manuscript (P19). Still, we do not expect a general solution to the QMC sign problem (see second paragraph of the manuscript, P2). We feel that further discussion would not be appropriate in the present context, but rather believe that it remains an interesting topic for further research how far the general idea of QMC in a cluster basis can be pushed.
We also do not quite understand the second "weakness" (point 2)). Figures 5-12 are intentionally formatted in the same way, but they still represent some carefully selected examples in order to
i) show that we can indeed perform efficient QMC simulations for the thermodynamic behavior of the frustrated ladder throughout the entire phase diagram,
ii) illustrate different physical cases, and compare situations with and without local conservation laws (Figs. 7 and 8 versus Figs. 5 and 6). Consequently, Figs. 5, 6, 9, 10, and 12 also each contain different curves to analyze the physics behind the numerical data.
Furthermore, Fig. 9 constitutes the only example in our manuscript where the ED results are indistinguishable from QMC. In the specific heat shown in Figs. 6(a) and 8(a), ED exhibits qualitative artifacts while in Figs. 5, 7, and 10 quantitative deviations are still visible, most notably in the specific heat, i.e., panel (a). Thus, QMC is indeed essential to obtain quantitatively reliable results for the thermodynamic quantities, in particular upon approaching the phase transition.

---

## Round 1 · Referee Report · Anonymous · 2017-5-22

Strengths
1. The manuscript is well written, presenting the clear explanation of their motivation and analysis.
2. It is of interest to readers that the unbiased Monte Carlo calculation is feasible in the whole parameter region of the frustrated spin-ladder model.
3. The development of the present method would be helpful for comparison to experimental data, especially in physical quantities at finite temperatures, which are not easy to calculate in other methods, such as the DMRG method.
Weaknesses
1. On the flip side, although the present results are interesting, their analysis on the possible continuous phase transition is far from conclusive.
Report
This manuscript is studying the average sign of configurations in the rung-basis quantum Monte Carlo simulation for the frustrated spin-ladder system. The authors report that the sign problem is mild in the whole parameter region of the system, while it is severe in use of the standard single-site basis. By calculating the specific heat and the uniform susceptibility, they study the ground state and the excitation of the frustrated spin ladder. Interestingly, the excitation gap is very small and may be zero at the estimated phase transition point in the weakly coupled chains. This result suggests that the phase transition may be a continuous phase transition, which is consistent with the previous analyses.
Requested changes
1. It is mentioned in the manuscript that configurations on the open boundary condition always have the positive sign. The authors should explain why they need to use the periodic boundary condition and study the average sign.
2. I fail to understand the reason of the minimum in the average sign as a function of temperature regarding Fig. 4. The authors explain it is because the configurations with the negative sign, which has an operator string wrapping around the entire system, never constitute a macroscopic fraction of all configurations. This is not clear to me. In addition, why does the temperature at which the average sign takes a minimum decrease with the system size?
3. In Fig. 12, the authors claim that the susceptibility of the frustrated ladder approaches a logarithmic singularity in a similar way to the spin chain. It is, however, not seen clearly in the numerical data. It is required to simulate a larger system at lower temperature in order to show the logarithmic singularity in the susceptibility.
4. Also regarding Fig. 12, the explanation of the susceptibility reduction compared to the decoupled chains is not clear. What does "the matrix-element effect to this coupling" mean? Does the word here "this coupling" mean the rung-coupling perturbation or the spinon confinement? How does the sensitivity reduce the susceptibility while not affecting the specific heat much?
Author: Andreas Honecker on 2017-05-31 [id 139]
(in reply to Report 2 on 2017-05-22)
We would like to thank the referee for her or his careful reading of the manuscript and for the overall positive evaluation of our work.
The referee suggests four changes.
-
The referee asks why, if there are no sign problems in the system with open boundary conditions (OBCs), we use periodic boundary conditions (PBCs) at all. In simulations on finite systems, OBCs are known to give rise to additional surface terms and thus require larger system sizes to ensure convergence to the thermodynamic limit than do PBCs. Because the sign problem for PBCs costs us an additional 25% in CPU time in the worst case (as noted on P10, in the last paragraph of section 3.3), it would therefore be significantly more expensive to use OBCs for all of our simulations. We agree that it is helpful to add a more explicit statement to this effect in the closing paragraph of section 3.3.
-
The referee may not have understood the meaning of our statements about the average sign. We do not purport to offer a full explanation for all of its behaviour. The best we can achieve within the scope of the present paper is to state our numerical results as accurately as possible in order to frame the open questions for subsequent research. The behaviour of the average sign in the rung basis is indeed highly unusual. Empirically, the average sign is due to topologically non-trivial configurations, i.e. non-zero winding around the boundary. This has three consequences: i) the average sign is sensitive to the correlation length, which suggests that it is a cheap observable for detecting a second-order phase transition. This is in fact another reason for using PBCs (item 1. above). ii) the "sign problem" improves with increasing system size while in a standard situation it becomes exponentially bad with system size. iii) the "sign problem" is "healed" at low temperatures, while again in standard situations it becomes exponentially bad at low temperatures. We have invested some thought into the meaning of these results; the discussion of section 3.2 of the manuscript is the fullest extent of the analytical discussion that we have managed. Because we do not wish to speculate, our statements beyond this level remain largely numerical observations (as, for example, in section 3.3). Deeper questions concerning the nature and evolution of the sign problem, as well as whether one may prove rigorously that it can be eliminated in an optimised basis, certainly rank as key problems for further research.
-
We are grateful for this remark and apologise for not expressing this point sufficiently clearly. Indeed, verification of logarithmic behaviour remains difficult even if an "exact" Bethe-Ansatz solution is available (our Refs. [82,83]). However, while our QMC data constitute no proof of a logarithmic singularity, we find the similarity in shape of the susceptibility curve to the well-understood case of the spin-1/2 Heisenberg chain to be highly suggestive and that it is helpful to point this out for the understanding of the reader. For clarity we suggest that the phrase "... approaches a logarithmic singularity ..." in section 4.5 be removed in its first appearance, where the data are described, and restored in the following paragraph, where the interpretation in terms of nearly-decoupled chains is offered.
-
On the referee's first point, We apologise for the complexity of this sentence. However, because "this coupling" was specified explicitly 10 words earlier in the same sentence, we were unable to find any grounds for ambiguity. We suggest that the last two phrases of this sentence be permuted to give the combination "... and in part the sensitivity to this coupling of matrix-element effects." On the referee's second point, we do not know at a fundamental level why the specific heat is affected so little, when compared to the susceptibility, by the presence of the interchain coupling. Our sentences on this point are intended to summarise our numerical results, to frame the question and to offer one plausible direction (namely matrix-element effects) in which future research may look for a deeper understanding.

---

## Round 1 · Referee Report · Anonymous · 2017-5-29

Strengths
1) the paper provides a pedagogical discussion of a promising route to deal with sign problems in quantum monte carlo simulations
2) the discussion is based on a paradigmatic frustrated ladder Heisenberg system
3) the authors offer an intuitive motivation why the sign problem is significantly reduced in their model by switching to a rung basis
Weaknesses
1) only a discussion of the parameter regime is given, where the sign problem is significantly reduced; for example, there is no quantitative discussion of the effect of broken leg symmetry on the extent of the sign problem which is only left as an outlook.
2) Figs. 5-11 appear somewhat repetitive (also, in about half of the panels in Figs 5-10, for given model system, ED already gives all features quantitatively correct)
Report
The paper is well polished and pedagogical.
Requested changes
It would have been great to incorporate data and discussion on broken leg symmetry; specifically, the rung-basis in terms of symmetry triplet and antisymmetric singlet is natural if the Hamiltonian is symmetric under leg exchange. If this symmetry is broken say on an intermediate level, how does this effect the QMC sign? what would be a good choice of cluster / basis in this case?

---

## Round 2 · Author Response

Minor modifications in sections 3.3 and 4.5.

---

## Round 2 · List of Changes

1) Comment on the use of periodic boundary conditions added at the end of section 3.3 (P10). 2) Remark on "logarithmic singularity" in the low-temperature limit of the susceptibility in Fig. 12(b) moved (section 4.5, P17). 3) "matrix-element effects" moved to a different position in the sentence (section 4.5, P17).

---

## Editorial Decision

published